# MarCos: Deep Thinking by Markov Chain of Continuous Thoughts

## Abstract

The current paradigm for reasoning in large language models (LLMs) involves models "thinking out loud" via a sequence of tokens, known as chain-of-thought (CoT). This approach, while effective, has several significant drawbacks. Firstly, inference requires autoregressive generation of often thousands of CoT tokens, which is slow and computationally expensive. Secondly, it constrains reasoning to the discrete space of tokens, creating an information bottleneck across reasoning steps. Thirdly, it fundamentally entangles reasoning with token generation, forcing LLMs to "think while speaking," which causes potentially short-sighted reasoning. In light of these limitations, we re-imagine reasoning in LLMs and present a new paradigm: **MarCos**. In our approach, rather than autoregressively generating tokens, we model reasoning as a hidden Markov chain of continuous, high-dimensional "thoughts". Each reasoning step involves a transition of the internal thoughts, where explicit reasoning steps (which may consist of hundreds of tokens) serve as observable variables, which are windows to peek into the implicit thoughts. Since this latent process is incompatible with the standard supervised learning, we further propose a two-phase variational training scheme. Our experiments on three benchmarks demonstrate that MarCos outperforms existing continuous reasoning methods and, for the first time, achieves performance comparable to token-based CoT, even surpassing it by 4.7% on GSM8K with up to $15.7\times$ speedup in inference. Beyond this, MarCos offers additional advantages, such as step-level instead of token-level control over randomness, opening significant opportunities for reinforcement learning and reasoning in LLMs.

## 1 Introduction

large language models (LLMs) have been shown to benefit greatly from the ability to generate chain-of-thoughts (CoT) in natural language (Wei et al., 2022; Xu et al., 2025). Despite its simplicity and applicability, CoT's causal style can lead to very slow inference, since tokens must be generated one by one. Meanwhile, it forces the model to think and speak simultaneously. In other words, the model needs to generate one token immediately after a single forward computation, leaving little room for thoughtful planning beforehand (Levelt, 1993; Dell & O'Seaghdha, 1992). Besides, the discretization at each time step heavily restricts the information bandwidth between time steps. For example, information at the last transformer layer can only be passed to the next step through the generated tokens, which leads to severe information loss.

To mitigate the previous issues of token-by-token reasoning, recent advances (Deng et al., 2023; 2024) find that it is possible to make models think in a continuous space. As a representative example, Coconut (Hao et al., 2024) replaces discrete tokens in LLMs' output with continuous vectors, which allows higher information bandwidth between time steps. CoLaR (Tan et al., 2025) further adds an extra loss term to align the continuous vectors with a representation of discrete token blocks in ground-truth reasoning paths. Although these methods reduce the number of time steps required to reach a final answer, they mainly work by compressing multiple discrete tokens into a continuous vector, which means that their latent thinking essentially remains a form of token-based decision-making. Moreover, most of them adopt a static transition function between continuous vectors during reasoning, which is unsuitable for learning the diverse and multimodal solution space.

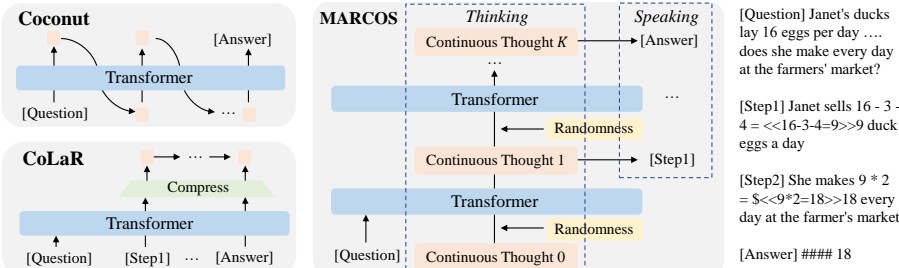

Figure 1: Comparison of MARCOS and two representative continuous reasoning methods, Coconut and CoLaR. In MARCOS, the *thinking* process is modeled as iterative transitions of continuous thoughts (0 to $K$) with incorporated randomness. The *speaking* process (optionally) translates these thoughts into natural language (e.g., [Step1] to [Answer]).

In this paper, we propose **MARCOS**, a new paradigm that models the reasoning process as a Markov chain of continuous thoughts. As shown in Figure 1, we disentangle the internal *thinking* process from the decoding (*speaking*) process, using separate modules to handle each of them. Similar to the human reasoning process, speaking now becomes optional during thinking. This allows the model to fully deliberate before "speaking", aligning with neuroscience discovery that language is primarily a tool for communication rather than thought (Fedorenko et al., 2024; Fedorenko & Varley, 2016). In addition, by adopting continuous representations (which we refer to as neurons) of internal thoughts within the chain, we enable high-bandwidth information flow between thinking steps. To model the diverse distribution of possible reasoning paths, we further incorporate variational modules that explicitly control the randomness for each reasoning step. Compared with traditional LLMs, which handle randomness at token level during sampling, our step-level control makes it easier to steer the direction of reasoning. With this brand-new design, MARCOS not only makes reasoning significantly faster, but also introduces more suitable inductive biases that enhance reasoning quality.

We evaluate our MARCOS on three benchmarks—GSM8K (Cobbe et al., 2021), SVAMP (Patel et al., 2021), and MultiArith (Roy & Roth, 2015). The preliminary results show that our model substantially surpasses all continuous reasoning baselines. Moreover, **MARCOS is the first latent reasoning method that obtains performance on par with or even superior to token-based CoT reasoning**, with a $4.7\%$ accuracy improvement on GSM8K and up to $15.7\times$ inference-time speedup. Beyond accuracy, our analysis reveals the functional independence between thinking neurons and variational modules, and uncovers that different randomness dimensions control different speaking properties such as sentence length and step depth. This opens up the possibility of manually controlling reasoning behaviors. Furthermore, we show that our model remains competitive when applying non-autoregressive (NAR) decoding in the speaking stage, indicating that thinking and speaking play well-separated roles in our framework. In summary, our contributions in this paper are:

- We propose MARCOS, a new paradigm that models reasoning as a Markov chain of continuous thoughts and operates in decoupled *thinking-speaking* stages.

- We introduce a principled way to model the randomness in reasoning, which achieves two key advances: (1) enabling explicit control over the reasoning process, and (2) demonstrating the feasibility of pre-determining step-level randomness before token generation.

- Experiments on three benchmarks show that MARCOS achieves state-of-the-art reasoning performance, improving over the best continuous reasoning model by $8.66\%$ accuracy and even surpassing token-based CoT by $4.7\%$ on GSM8K with up to $15.7\times$ speedup.

## 2 RELATED WORK

Various continuous reasoning methods have emerged to mitigate the issues caused by token-by-token reasoning, which can be broadly categorized into time-axis and depth-axis approaches.

**Time-axis latent reasoning.** Most time-axis continuous reasoning approaches aim to improve the inference efficiency by compressing the discrete tokens in explicit CoT into a few continuous steps. For example, iCoT-KD (Deng et al., 2023; 2024) distills entire reasoning paths into single inference

steps (a single time step in LLM inference) before generating the final answer, while curriculum learning is adopted in iCoT-SI (Deng et al., 2024) to gradually encourage the model to shift from explicit rationales toward fully latent reasoning. However, compressing whole CoT reasoning paths, which may consist of thousands of tokens, into a single inference step, is too aggressive and degrades model performance. Consequently, more recent works adopt intermediate continuous vectors, each representing a block of consecutive discrete tokens in CoT reasoning. For example, CODI (Shen et al., 2025) introduces an autoregressive latent variable framework that is trained by aligning its hidden states at the last step before the final answer with an explicit CoT model. Coconut (Hao et al., 2024) progressively replaces blocks of discrete word tokens with continuous vectors and concatenates them together in the reasoning chain. CoLaR (Tan et al., 2025) further adds supervision to align the latent states with embeddings of token blocks that they are trying to replace in explicit CoT. While improving inference efficiency, these methods essentially remain a form of token compression and could still be restricted by token-level reasoning. Moreover, their continuous vectors are updated either in a fixed manner or via simple Gaussian transitions, which limits their ability to model the diversity of reasoning and exploratory thinking.

**Depth-axis latent reasoning.** These approaches modify the transformer architecture to support latent reasoning. One representative is the Looped Transformer (Giannou et al., 2023; Yang et al., 2024), which introduces recurrence within transformer layers, reusing the same parameters across iterations to progressively deepen internal deliberation. Similarly, Geiping et al. (2025) apply transformer blocks recurrently for each token. It allows the model to unroll to arbitrary depth, enabling test-time scaling up without generating additional tokens. Compared with MARCOS, these methods lack supervision for intermediate reasoning states. Consequently, they can be simply seen as parameter-efficient ways to increase the depths of transformers, which are orthogonal to our focus.

## 3 MARCOS: MARKOV CHAIN OF CONTINUOUS THOUGHTS

In this section, we propose MARCOS, a new paradigm for latent thinking in continuous space. Specifically, we model the thinking process as a conditional hidden Markov model (cHMM), where the input questions are the conditions, the internal thoughts are hidden variables, and spoken sentences are observable variables. This formulation enforces a clear separation between thinking and speaking: thinking corresponds to transitions of latent thoughts, and speaking is the emission process from latent thoughts to natural language. Crucially, latent thought at each time step represents a *reasoning step*, which enables the model to plan thoroughly before producing a conclusion. Compared with traditional token-based CoT, our paradigm offers two key advantages: (1) by disentangling thinking from speaking, it enables more structured and deliberate reasoning beyond token-by-token generation; and (2) the entire thinking process is carried out in continuous space, thereby avoiding the information loss caused by token discretization.

The transition functions between latent thoughts are typically static mappings in most previous work on latent reasoning. However, this design is suboptimal, since even for a simple math problem, there may be multiple plausible reasoning paths. As a result, static transitions are insufficient to model the diverse distribution of possible thoughts, which limits the model's exploration during inference.

To tackle this problem, MARCOS maps each thought to a multimodal distribution of possible thoughts at the next step, rather than a single deterministic one. By doing so, we can explicitly model the randomness in the thinking process without relying on token-level sampling. We will first introduce this basic structure in Section 3.1. However, the stochastic mapping poses a challenge in training: under the complicated transition, the probability of a ground-truth step becomes intractable to maximize directly. To address this, inspired by VAE (Kingma & Welling, 2013), we propose a two-phase training scheme that infers the posterior distribution of thoughts given the ground-truth reasoning step and optimizes an ELBO-like objective. To simplify training and inject inductive biases for more interpretable control over randomness, we further incorporate ideas from sparse autoencoders (Cunningham et al., 2023), which we will elaborate on in Section 3.2.

### 3.1 MODEL STRUCTURE

This section explains how MARCOS "thinks" and generates solutions to a question during inference. It consists of three separate stages: understanding, thinking, and speaking, which provide conditions,

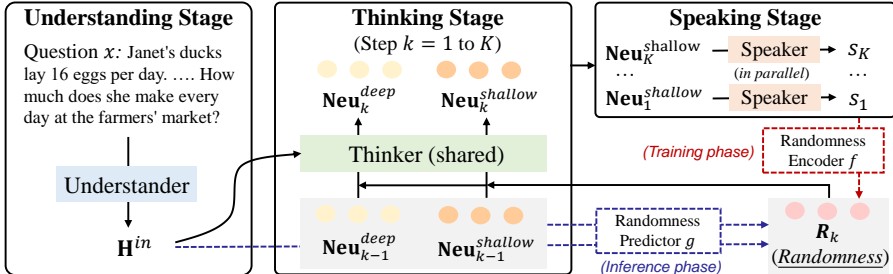

Figure 2: Illustration of our MARCOS model. In the thinking stage, MARCOS continuously reasons in the latent space for $K$ steps by updating neurons $\mathbf{Neu}_k^{deep}$ and $\mathbf{Neu}_k^{shallow}$. In the speaking stage, $\{\mathbf{Neu}_k^{shallow}, k = 1, ..., K\}$ are translated to natural language in parallel. If intermediate steps are not required, we only need to translate $\mathbf{Neu}_K^{shallow}$ to a final answer. During training, the random variable $\boldsymbol{R}_k$ is derived from the ground-truth sentence $\boldsymbol{s}_k$ using the randomness encoder $f$, while in inference, it is predicted by the randomness predictor $g$ based on $\mathbf{Neu}_k^{deep}$, $\mathbf{Neu}_k^{shallow}$, and $\mathrm{H}^{in}$.

transition probabilities, and emission probabilities for the conditional hidden Markov model, as shown in Figure 2.

**Understanding.** Firstly, given the input question $x$ of length $n$, we use a transformer-based understander to convert it into a sequence of feature vectors $\mathrm{H}^{in} = \{\boldsymbol{h}_1, ..., \boldsymbol{h}_n\}$:

$$\mathrm{H}^{in} = \text{Understander}(x), \tag{1}$$

where $\mathrm{H}^{in}$ can be seen as an $n \times d$ real matrix, and $d$ is the latent dimension.

**Thinking.** Then, as the hidden variable of the cHMM, our continuous thought is realized by a set of real-valued neurons. Specifically, inspired by neuroscience evidence of functional specialization in the human brain (Kanwisher, 2010), we explicitly split them into two groups: $\mathbf{Neu}^{deep} \in \mathbb{R}^{T \times d}$ and $\mathbf{Neu}^{shallow} \in \mathbb{R}^{S \times d}$, with the intuition that $\mathbf{Neu}^{deep}$ facilitates deeper reasoning, while $\mathbf{Neu}^{shallow}$ represents shallower and more readily verbalizable reasoning, as it interacts more closely with the speaking module (see Eq. 4). $T$ and $S$ denote the numbers of neurons in each group. Each thinking step can be viewed as updating both $\mathbf{Neu}^{deep}$ and $\mathbf{Neu}^{shallow}$, conditioned on the input feature $\mathrm{H}^{in}$ and their own previous states.

As described earlier, a key difference between MARCOS and existing continuous reasoning approaches is that it explicitly models the randomness of the thinking process. Concretely, MARCOS maps the current thought to a distribution of possible thoughts at the next step. However, due to the multi-modality of thought distributions, it is insufficient to directly parameterize them as simple distributions such as Gaussians. We address this problem by introducing an auxiliary random variable $\boldsymbol{R}_k \in \mathbb{R}^{\tau \times d}$ at each step $k$, where $\tau$ is a scaling factor. Instead of directly mapping the current thought to the multimodal and high-dimensional distribution of thoughts at the next step, we first map it to a distribution of $\boldsymbol{R}_k$, which lies in a better-behaved, lower-dimensional space. This separation facilitates more controllable randomness. Specifically, in Section 3.2, we point out that by adding a sparsity constraint to $\boldsymbol{R}_k$, we could encourage each dimension of $\boldsymbol{R}_k$ to represent a single random factor and stabilize the training process. In Section 4.4, we observe that $\boldsymbol{R}_k$ effectively controls different aspects of thinking, including high-level factors (e.g., searching directions, reasoning depth) and low-level choices (e.g., expression format, sentence length).

Since $\boldsymbol{R}_k$ varies with input information and the latent thought at step $k$, we employ a learnable randomness predictor $g$ to model its distribution:

$$\boldsymbol{\mu}_k, \boldsymbol{\sigma}_k = g(\mathbf{Neu}_{k-1}^{deep}, \mathbf{Neu}_{k-1}^{shallow}, \mathrm{H}^{in}). \tag{2}$$

A sample of $\boldsymbol{R}_k \sim \mathcal{N}(\boldsymbol{\mu}_k, \boldsymbol{\sigma}_k)$ is then drawn to guide the transition from the current thought to the next one:

$$\mathbf{Neu}_k^{deep}, \mathbf{Neu}_k^{shallow} = \text{Thinker}(\mathbf{Neu}_{k-1}^{deep}, \mathbf{Neu}_{k-1}^{shallow}, \mathrm{H}^{in}, \boldsymbol{R}_k). \tag{3}$$

To facilitate this information flow, the thinker is implemented as a bi-directional transformer. Through multiple layers of cross-attention and feed-forward computation, it iteratively updates the thinking neurons. The initial values of neurons, $\mathbf{Neu}_0^{deep}, \mathbf{Neu}_0^{shallow}$, are learnable parameters.

**Speaking.** After each thinking step, some thoughts in $\mathbf{Neu}_k^{shallow}$, such as plans, operations, and intermediate conclusions, may become ready for verbalization. We employ a transformer-based speaker to translate them into the discrete space of sentences:

$$s_k = \text{Speaker}(\mathbf{Neu}_k^{shallow}). \tag{4}$$

Different from traditional LLMs, thinking in MARCOS does not depend on speaking, so our speaking stage is optional for all intermediate steps and can be skipped except for the final step when only an answer is required. Moreover, the speaking at each step is independent, each conditional on its corresponding neurons, so they can run in parallel for better inference efficiency.

In addition, unlike traditional LLMs where randomness is implemented through token-level sampling, randomness in MARCOS is resolved at step level in the thinking stage *before* the speaker generates specific sentences. The effectiveness of this architecture indicates that complex randomness can be simulated "in one pass" rather than through multiple "iterative" steps. This provides a new perspective for the development of non-autoregressive (NAR) language models with one-pass generation, which may provide a direction to accelerate diffusion LLMs (Gulrajani & Hashimoto, 2023; Sahoo et al., 2024; Li et al., 2025). Besides, since the speaker mainly serves as a deterministic translator from $\mathbf{Neu}_k^{shallow}$ to explicit sentences, it is also suitable for NAR decoding, which can further accelerate the speaking stage. As shown in Section 4.4, even the simplest NAR decoding strategy still obtains competitive reasoning accuracy. This highlights that our model achieves a clear separation between thinking and speaking, ensuring that the reasoning quality remains stable while allowing flexibility in the choice of speaking strategy.

## 3.2 TRAINING

In this part, we discuss how to effectively train MARCOS, with a focus on teacher-forcing training. Assume we have a training sample that consists of a question $x$ and its step-by-step solution $y = \{y_1, ..., y_N\}$. Similar to traditional LLMs, the training objective is to find the parameters that maximize the probability of generating all $y_k$. However, because sampling occurs in thinking rather than in speaking, we no longer have a closed-form conditional probability for $y_k$.

To solve this, we borrow the idea of evidence lower bound (ELBO) used in training VAEs to maximize the marginal probability of $y_k$ at each step. Specifically, a general ELBO can be written as:

$$\mathcal{L}_k^{\text{ELBO}} = \mathbb{E}_{q(\mathbf{R}_k|y_k)}\big[\log p(y_k \mid \mathbf{R}_k)\big] - \text{KL}\big(q(\mathbf{R}_k \mid y_k) \,\|\, p(\mathbf{R}_k)\big), \tag{5}$$

which is a combination of a reconstruction term and a KL term. In our case, the VAE encoder $q$ is realized by a learnable function $f$ (referred to as randomness encoder in Figure 2) to map $y_k$ to the posterior mean and variance of $\mathbf{R}_k$:

$$\hat{\boldsymbol{\mu}}_k, \hat{\boldsymbol{\sigma}}_k = f(y_k). \tag{6}$$

The VAE decoding is simply a step of thinking and speaking in Eqs. 3 and 4, which maps $\mathbf{R}_k$ back to the sentence space. Then the reconstruction loss can be expanded as:

$$\mathcal{L}_k^{re} = \mathbb{E}_{\mathbf{R}_k \sim \mathcal{N}(f(y_k))}\big[\log p(s_k = y_k | \mathbf{Neu}_{k-1}^{deep}, \mathbf{Neu}_{k-1}^{shallow}, \mathbf{H}^{in}, \mathbf{R}_k)\big]. \tag{7}$$

Unlike vanilla VAEs with fixed isotropic Gaussian priors, our randomness predictor $g$ in Eq. 2 provides a different prior for different questions and latent thoughts as described in Section 3.1. Consequently, to minimize the KL divergence between the prior (Eq. 2) and the posterior (Eq. 6), we have:

$$\mathcal{L}_k^{KL} = \text{KL}\big(\mathcal{N}(g(\mathbf{Neu}_{k-1}^{deep}, \mathbf{Neu}_{k-1}^{shallow}, \mathbf{H}^{in})) \,\|\, \mathcal{N}(f(y_k))\big). \tag{8}$$

In addition, to make the space of $\mathbf{R}_k$ simpler and easier to interpret, we introduce an extra sparsity loss $\mathcal{L}_k^{sparse} = \|\mathbf{R}_k\|_1$. The motivation is that at each thinking step, only a small set of dimensions need to be active. For example, dimensions that are useful for breadth-first reasoning may not be activated during depth-first reasoning. To reflect this inductive bias and encourage each dimension of $\mathbf{R}_k$ to capture a "mono-semantic" factor in thinking, we draw inspiration from sparse autoencoders and implement the function $f$ as a $d$-dimensional transformer followed by an MLP that expands the dimension from $d$ to $\tau \times d$. The overall training loss of MARCOS is:

$$\mathcal{L} = \sum_{k \in \{1,2,...,N\}} \big[-\mathcal{L}_k^{re} + \mathcal{L}_k^{KL} + \lambda\mathcal{L}_k^{sparse}\big], \tag{9}$$

where $\lambda$ is a hyperparameter that controls the strength of the sparsity loss. However, in $\mathcal{L}_k^{KL}$, both the posterior and the prior are learnable, which can cause training instability. To address this, we adopt a two-phase training scheme, where we first minimize the other two terms in Eq. 9 and then train $g$ to minimize $\mathcal{L}_k^{KL}$ with all other parameters fixed.

# 4 EXPERIMENTS

## 4.1 EXPERIMENTAL SETUP

**Datasets and Tasks.** Following Tan et al. (2025), we train our model on **GSM8K-Aug** dataset (Deng et al., 2023), an augmented version of **GSM8K** (Cobbe et al., 2021), which contains 385K training samples. Then, we evaluate on the original GSM8K test set (in-domain task), as well as two out-of-domain benchmarks: (1) **SVAMP** (Patel et al., 2021), consisting of 4,138 arithmetic problems constructed by subtle variations in wording and semantic perturbations, which aims to evaluate the robustness of reasoning ability, (2) **MultiArith** (Roy & Roth, 2015), a collection of 600 problems from MAWPS (Koncel-Kedziorski et al., 2016) that require multi-step reasoning. Our evaluation metrics include (1) Accuracy (Acc), which reflects the reasoning ability; (2) Training Time (Train), which measures the time cost for model training; and (3) Test Time (Test), which measures inference efficiency by counting the total time required to produce solutions for all test samples. More detailed descriptions of the evaluation setup are provided in Appendix A.

**Baselines.** We compare with (1) **CoT-SFT** (Wei et al., 2022), which finetunes models on solution-based data to elicit structured and logical reasoning, (2) **iCoT-SI** (Deng et al., 2024), which gradually removes explicit reasoning steps during training to encourage implicit reasoning, (3) **Coconut** (Hao et al., 2024), which treats latent representations as a special token and inputs them alongside word tokens into an autoregressive model, (4) **CoLaR** (Tan et al., 2025), which compresses reasoning tokens and learns to predict the distribution of these compressed embeddings, (5) **CODI** (Shen et al., 2025), which self-distills hidden activations across all layers at the selected distillation token.

**Implementation Details.** We use the same transformer structure as Qwen2.5-0.5B (Team, 2024) for our understander, thinker, speaker, and randomness encoder, while the randomness predictor is a simple two-layer MLP. Therefore, the total parameter count of MARCOS is approximately $0.5\text{B} \times 4 = 2\text{B}$. For fair comparison, we implement baseline models with both 0.5B and 3B parameters, with the same transformer structure as Qwen2.5. For continuous reasoning baselines, we follow their original recipes to first train an explicit CoT for initialization and then finetune with only structured equations. For our model and CoT-SFT, except for an equation-based version for direct comparison, we also train a text-based version, which is closer to real-world applications.

Since our model is a fundamentally new reasoning paradigm, it is not directly compatible with existing token-by-token pretraining. Moreover, due to limited computational resources, we are unable to pretrain on the same scale of data as Qwen2.5. Therefore, **we train our model from scratch**. For fair comparison, all baselines are also trained from scratch, allowing a direct comparison without influence of existing pretraining or finetuning. More details of experimental settings are provided in Appendix A. In addition, we discuss potential directions for pretraining our method in Appendix D. Our code is available at `https://anonymous.4open.science/r/MarCos/`.

## 4.2 MAIN RESULTS

As shown in Table 1, MARCOS significantly outperforms existing baselines in accuracy, while providing up to $15.7\times$ speedup compared with token-based CoTs. Specifically, MARCOS (equation) achieves a $8.66\%$ accuracy improvement over the best CoLaR baseline on GSM8K with Qwen2.5-0.5B and even beats its 3B version by $1.51\%$. Beyond in-domain performance, MARCOS yields gains of up to $4.68\%$ on SVAMP and MultiArith, which demonstrates strong out-of-domain generalization. Notably, MARCOS also exceeds CoT-SFT in most settings, which to the best of our knowledge, provides preliminary evidence that implicit reasoning can reach the level of explicit CoT. In particular, when trained with text supervision, MARCOS improves accuracy by $4.7\%$ on GSM8K. These results highlight its ability to execute implicit and deep reasoning effectively.

In terms of efficiency, MARCOS not only outperforms token-based CoTs, but also achieves faster inference than many of the 0.5B baselines. This underscores that modeling reasoning as a Markov

Table 1: Results on GSM8K, SVAMP, and MultiArith with Qwen2.5 backbones (0.5B and 3B). All continuous reasoning baselines are trained on equation data, following their original papers. We report the best implicit models in bold and the runner-ups with underline.

| Model | Train (h) | GSM8K | | SVAMP | | MultiArith | |
| --- | --- | --- | --- | --- | --- | --- | --- |
| | | Acc (%) | Test (s) | Acc (%) | Test (s) | Acc (%) | Test (s) |
| *Training Data: Text* | | | | | | | |
| CoT-SFT (0.5B) | 0.31±0.01 | 13.27±0.59 | 80.95±5.78 | 23.25±0.39 | 216.56±11.88 | 25.20±0.47 | 18.32±1.81 |
| CoT-SFT (3B) | 1.81±0.00 | 12.66±0.20 | 177.37±1.91 | 24.24±0.50 | 537.78±12.14 | 23.44±1.40 | 58.20±2.50 |
| MARCOS | 1.13±0.01 | **17.97**±0.96 | 17.53±0.29 | **24.39**±1.10 | 34.14±0.49 | **23.67**±0.53 | 5.90±0.17 |
| $- w/o$ **Neu**$^{deep}$ | 1.12±0.01 | 16.91±1.60 | 17.28±0.13 | 22.04±0.29 | 34.83±0.26 | 20.28±0.25 | 5.68±0.16 |
| $- w/o$ $R_k$ | 0.77±0.01 | 16.07±0.00 | 17.20±0.10 | 20.27±0.00 | 32.43±0.11 | 16.66±0.00 | 5.65±0.12 |
| $- w/o$ $sparsity$ | 1.14±0.01 | 1.16±0.09 | 16.99±0.10 | 2.16±0.09 | 32.35±2.01 | 0.78±0.09 | 6.15±0.21 |
| *Training Data: Equation* | | | | | | | |
| CoT-SFT (0.5B) | 0.38±0.01 | 20.31±0.85 | 32.97±1.03 | 27.92±0.36 | 162.12±7.43 | 41.48±0.50 | 9.61±0.91 |
| iCoT-SI (0.5B) | 0.43±0.03 | 14.81±0.55 | 13.11±1.62 | 18.46±0.54 | 17.30±0.31 | 21.33±0.43 | 2.32±0.05 |
| Coconut (0.5B) | 1.50±0.25 | 15.27±0.05 | 59.53±0.53 | 17.76±0.01 | 106.74±1.46 | 15.50±0.01 | 15.98±0.28 |
| CoLaR (0.5B) | 1.17±0.00 | 15.45±0.42 | 17.20±0.45 | 23.09±0.09 | 35.80±0.45 | 38.60±0.55 | 3.80±0.45 |
| CODI (0.5B) | 13.95±0.09 | 2.55±0.17 | 8.58±0.25 | 1.66±0.22 | 24.88±0.26 | 1.60±0.25 | 3.48±0.24 |
| CoT-SFT (3B) | 1.23±0.00 | 22.70±0.36 | 54.07±2.16 | 27.25±0.46 | 125.47±7.67 | 50.20±1.60 | 10.73±0.17 |
| iCoT-SI (3B) | 2.40±0.10 | 14.08±0.67 | 13.23±0.06 | 16.26±0.27 | 29.02±0.44 | 19.67±0.58 | 3.49±0.05 |
| Coconut (3B) | 12.62±1.06 | 9.95±0.04 | 127.35±0.44 | 13.53±0.11 | 735.14±3.80 | 8.78±0.19 | 26.56±0.66 |
| CoLaR (3B) | 2.21±0.00 | 22.60±0.19 | 38.13±0.21 | 25.83±0.22 | 129.73±0.96 | 41.83±0.44 | 18.93±0.73 |
| CODI (3B) | 34.81±0.00 | 1.24±0.36 | 12.91±0.14 | 1.12±0.17 | 34.18±0.24 | 1.20±0.68 | 4.94±0.14 |
| MARCOS | 1.03±0.01 | **24.11**±0.97 | 17.76±0.04 | **27.77**±0.32 | 34.02±0.11 | **42.33**±0.56 | 6.02±0.28 |
| $- w/o$ **Neu**$^{deep}$ | 0.99±0.01 | 21.07±0.76 | 17.09±0.30 | 25.83±0.19 | 35.24±0.11 | 38.38±1.04 | 6.00±0.29 |
| $- w/o$ $R_k$ | 0.74±0.01 | 23.99±0.00 | 17.67±0.09 | 26.90±0.00 | 33.78±0.54 | 42.17±0.00 | 5.99±0.08 |
| $- w/o$ $sparsity$ | 1.01±0.02 | 1.16±0.10 | 17.00±0.21 | 1.33±0.05 | 27.63±2.08 | 0.56±0.10 | 4.96±2.57 |

chain and decoupling the thinking and speaking brings substantial computational advantages. Remarkably, at the same parameter scale, MARCOS requires the least training time, even less than most 0.5B continuous reasoning models. These findings demonstrate that our two-stage training scheme is both effective and exceptionally efficient.

### 4.3 ABLATION STUDY

To show the effectiveness of different components, we compare MARCOS with three of its variants: $w/o$ **Neu**$^{deep}$, which removes the deep thinking neurons; $w/o$ $R_k$, which removes the random variable; and $w/o$ $sparsity$, which removes the sparsity loss $L_k^{sparse}$.

In Table 1, we observe that removing any of these components leads to a substantial performance drop, and removing $L_k^{sparse}$ even leads to model collapse. This can be attributed to two reasons. On one hand, without the sparsity loss, the random variable $R_k$ degenerates into a dense vector, whose distribution may becomes too complex for the randomness predictor $g$ to learn in the second phase of training. On the other hand, without a proper control, the model may encode excessive semantic information of $y_k$ into $R_k$ via the randomness encoder $f$. This information can be easily merged into **Neu**$_k^{shallow}$ for the reconstruction of $y_k$, creating a training "shortcut", which prevents the model from learning genuine reasoning.

We also find that the relative importance of **Neu**$^{deep}$ and $R_k$ depends on the supervision type. When trained with equations, removing $R_k$ has less impact than **Neu**$^{deep}$, whereas with text supervision the opposite holds. This is because equations carry very little inherent randomness, while texts can exhibit more variability in aspects such as length and format, which will be validated in Section 4.4.

### 4.4 ANALYSIS AND DISCUSSIONS

**More thinking neurons, better performance?** We first investigate how the scale of thinking neurons and random variables influence the model's performance. Specifically, we vary the number of deep thinking neurons $T \in \{1, 5, 10, 20, 50\}$, shallow thinking neurons $S \in \{10, 50, 100, 200, 500\}$,

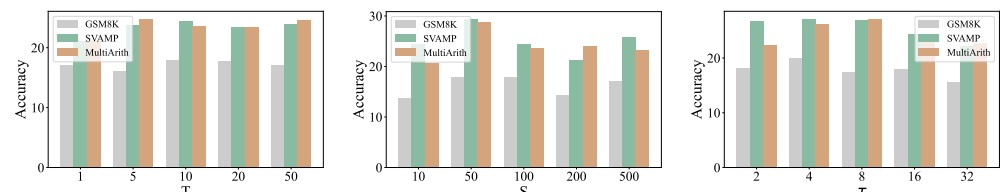

Figure 3: Performance of MARCOS (text) with different numbers of neurons and random variables.

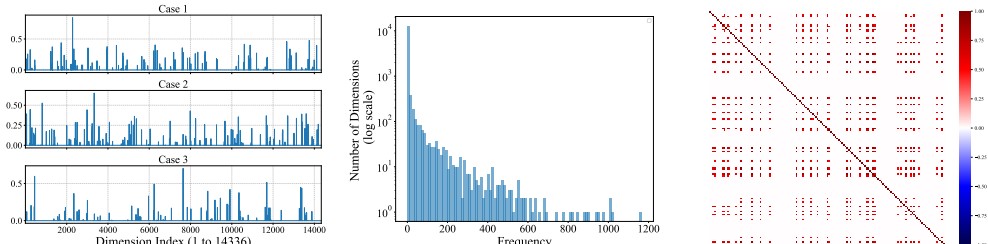

Figure 4: **Left**: $R_k$ of step 1 for three cases, **Middle**: Frequency distribution of all dimensions in $R_k$, **Right**: Correlation heatmap of the top 1% most frequently activated dimensions.

and random variables $\tau \in \{2, 4, 8, 16, 32\}$. Results for text-based and equation-based versions are reported in Figure 3 and Figure 7 in Appendix, respectively.

For both types of data, as the numbers of neurons $T$ and $S$ increase, the model performance exhibits an initial increase, followed by a period of stabilization or decline. This suggests that performance is initially constrained by the model's thinking capacity, but beyond a certain point, adding more neurons may introduce noise or lead to overfitting. We speculate that more neurons may be needed when the model is pretrained on massive integrated data, which we will investigate in future work.

Regarding the scale of randomness $\tau$, we observe different phenomena on text- and equation-based models. For the equation setup, a large $\tau$ generally leads to worse performance, consistent with our earlier hypothesis in Section 4.3 that equation supervision contains relatively little variability. In contrast, for the text setup, good performance is achieved only with a large $\tau$, indicating that a higher degree of stochasticity is required to adequately capture the diverse variations in natural language.

**How does $R_k$ control the randomness of reasoning?** As shown in Figure 4, different data samples activate different dimensions of random variable $R_k$. To quantify, we define dimensions with values greater than $0.1$ as *activated* and count activation frequencies across all steps for all samples. From the middle part of Figure 4, the distribution exhibits a long-tail pattern: while most dimensions are activated only once, a small subset of dimensions are activated at a very high frequency. This suggests that general randomness information may be controlled by these high-frequency dimensions, whereas case-specific randomness is reflected in the low-frequency ones.

To verify this, we select the top 1% most frequently activated dimensions and compute their pairwise Pearson correlation coefficients in Figure 4 (right). Interestingly, we find that several dimensions are highly correlated with each other. In Appendix B, we further visualize this correlation structure as a graph, where we observe a clique of four dimensions that may represent a form of "combinational control". Specifically, we identify that these dimensions mainly control two factors: (1) *output length*, i.e., the number of tokens in the reasoning steps, and (2) *output format*, i.e., whether the final answer is preceded by the marker "####". To illustrate, we manually intervene on these dimensions by setting

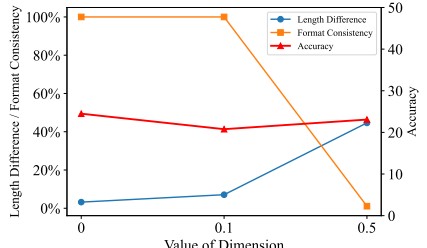

Figure 5: Results of intervening a clique of highly correlated dimensions in $R_k$.

their values to $\{0, 0.1, 0.5\}$ for all data samples. Figure 5 shows that with higher values, the generated solutions become substantially longer (blue line). For example, at $0.5$, the average length increases by $44.62\%$. Meanwhile, the strictness of format diminishes: at $0$ and $0.1$, almost all answers preserve the "####" prefix, whereas at $0.5$ the proportion drops sharply to $1.08\%$ (orange

| **Case 1:** Raphael went to buy some school supplies. He bought 4 pens which cost $1.5 each, 2 notebooks which cost $4 each, and a rim of bond paper which cost $20. How much did Raphael spend on everything? | **Original Solution:** *Step1:* Raphael spent 4 x $1.5 = $6 on pens. He spent 2 x $4 = $8 on notebooks. *Step2:* Therefore, he spent a total of $6 (for pens) + $8 (for pens) + $20 (for notebooks) = $34. *Step3:* #### 34. | $R_k(A) = 0.5$: *Step1:* Raphael spent 4 × $1.5 = $6 on pens He spent 2 × $4 = $8 on notebooks. In total, he spent $6 + $8 + $20 = $34. *Step2:* Therefore, he spent a total of $34. *Step3:* #### 34. | $R_k(B) = 0.1$: *Step1:* Raphael spent 4 pens × $1.5/pen = $6 on pens. He spent 2 notebooks × $4/notebook = $8 on notebooks. *Step2:* Therefore, he spent a total of $6 + $8 + $20 = $34 for the items. *Step3:* #### 34. |
| **Case 2:** Julie had $500. She spent 20% of it on clothes and then 25% of the remaining money on CDs. How much money did Julie have left? | **Original Solution:** *Step1:* Julie spent $500 * 20/100 = $100 on clothes. She had $500 - $100 = $400 left. *Step2:* She then spent $400 * 25/100 = $100 on CDs. So, she had $400 - $100 = $300 left. *Step3:* #### 300. | $R_k(C) = 0.5$: *Step1:* Julie had $500 * 20/100 = $100 left after buying clothes. She had $500 - $100 = $400 left after buying CDs. She then spent 25/100 * $400 = $100 on CDs. *Step2:* She had $400 and she spent $100 so she has 400-100 = $300 left. *Step3:* #### 300. | $R_k(D) = 0$: *Step1:* Julie spent $500 * 0.20 = $100 on clothes. She had $500 - $100 = $400 left. *Step2:* She then spent $400 * 25/100 = $100 on CDs. So, she had $400 - $100 = $300 left. *Step3:* #### 300. |

Figure 6: Case study of controlling four sets of dimensions (listed in Appendix C) in $R_k$ ($k = 1, ..., K$). For each case, the difference of model output after intervention is highlighted in red.

line). Crucially, despite these changes, reasoning accuracy remains largely unaffected (red line). These findings indicate that these dimensions independently control randomness factors such as verbosity and format, without encoding the essential reasoning content.

For the case-specific information, we perform a case-by-case analysis in Figure 6, where we highlight in red the differences between the controlled outputs and the original solutions. First, we observe that several dimensions affect the *reasoning depth*. For example, when setting the dimensions $A$ or $C$ to 0.5, both cases exhibit deeper reasoning at Step 1, where they complete the original Step 1 and Step 2 simultaneously. The difference between these two dimensions may arise because dimension $A$ primarily affects the final equation, whereas dimension $C$ influences the penultimate equation. This indicates that the stored information in random variables goes beyond surface-level linguistic style, and can in fact control diverse reasoning pathways. Second, we identify distinct functional roles of certain dimensions. In Case 1, dimensions $B$ control whether units are explicitly expressed in the equations (e.g., "4" vs. "4 pens"), whereas in Case 2, dimensions $D$ determine whether numbers are represented as fractions ("20/100") or decimals ("0.20"). These findings reflect that our variational module effectively captures nuanced variations in reasoning.

**Potential for non-autoregressive decoding in speaking stage.** A key advantage of MARCOS is the explicit separation between thinking and speaking. This means that a clear idea about what to say is already formed before entering the speaking stage. As a result, there will be less stochasticity and lower correlation between tokens during sentence decoding, which makes MARCOS well-suited for faster non-autoregressive generation. We verify this idea by directly training the speaker to generate 128 tokens in one pass. As shown in Table 2, even with such a naive training recipe, MARCOS still achieves performance comparable to autoregressive baselines such as iCoT-SI in Table 1. This finding highlights the potential of MARCOS to integrate more advanced NAR strategies, which is an important future direction.

Table 2: Reasoning accuracy of NAR decoding at the speaking stage.

| Method | GSM8K | SVAMP | MultiArith |
|---|---|---|---|
| MARCOS (text) | 9.02 | 14.96 | 8.33 |
| MARCOS (equation) | 14.25 | 22.82 | 30.12 |

## 5 CONCLUSION

In this work, we introduced MARCOS, a new paradigm that models reasoning of LLMs as a hidden Markov chain of continuous thoughts. Through extensive experiments, we showed that MARCOS not only surpasses existing continuous reasoning methods but also, for the first time, outperforms token-based CoT while providing substantial inference speedup. Looking ahead, our top priority is to verify the effectiveness of MARCOS when pretrained on massive corpus and develop dedicated reinforcement learning algorithms for the post-training of MARCOS. We will also explore more advanced NAR strategies to further improve the efficiency of MARCOS. For more discussions about this work's limitations and future directions, please refer to Appendix D.

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

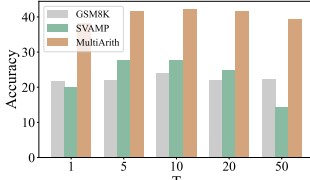 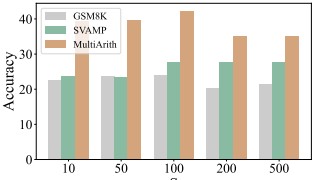 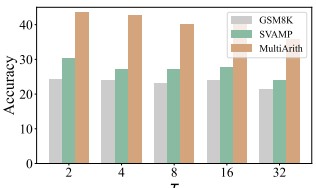

Figure 7: Performance of MARCOS (equation) with different numbers of neurons and random variables.

## A  DETAILS OF EVALUATION AND IMPLEMENTATION

In our experiments, Training Time is defined as the wall-clock time required to complete one epoch of training. To ensure a fair comparison, for baselines that require pre-training an explicit CoT model, the reported training time is the sum of (i) the time to train the explicit CoT model for one epoch, and (ii) the time to train their own model for one epoch. For our model, the reported training time is the sum of the two training phases, each measured over one epoch. Test Time is defined as the total time required to infer all test samples on a single GPU, using a batch size of 64. However, the original implementation of Coconut only supports inference with batch size = 1. We attempted to modify it to batch size = 64, but observed no significant improvement in efficiency. Therefore, for Coconut we report the time of its default setting, i.e., 4 GPUs with batch size = 1.

For our MARCOS, the number of neurons is set as $T = 10$, $S = 100$, and $\tau = 16$. It is trained by AdamW with a learning rate of $1e-4$ and a weight decay of 0.01. The training batch size is 256, and both training phases run for 10 epochs. To simplify prototyping, the number of thinking steps is fixed as $K = 3$, although our model also supports dynamically deciding this number. For text data, each period (".") is treated as a separate step. If a sample contains more than three steps, the original steps are evenly grouped and recombined into three steps. For fair comparison, the CoT-SFT baseline is trained for 20 epochs. All experiments are conducted on a server with 8 H200 GPUs.

## B  VISUALIZATION OF DIMENSION CORRELATION STRUCTURE

Figure 8 visualizes the correlation graph described in Section 4.4. In this graph, each node represents one of the top $1\%$ most frequently activated dimensions, and an edge is drawn between two nodes if their Pearson correlation exceeds 0.9. From the figure, we can clearly observe a clustered structure, suggesting that some dimensions tend to be co-activated in a highly consistent manner. Notably, within the cluster we further identify a clique formed by dimensions $\{2129, 3353, 6810, 7994\}$, which indicates that these dimensions jointly control aspects of general randomness.

## C  DIMENSION SETS FOR CASE STUDY

The dimension sets used in Figure 6 are: $A=\{$ 199, 3505, 5283, 7879, 9259, 9468, 9485, 11725$\}$, $B=\{4060\}$, $C=\{1379, 1583, 5475, 7017, 7074, 7521, 8683, 9635, 10460, 10539, 11716, 12012, 13337\}$, $D=\{1226\}$.

## D  LIMITATIONS AND FUTURE DIRECTIONS

The following are some limitations in this work, which also point to promising directions for future research. First, the experiments in this paper primarily focus on mathematical reasoning tasks. This choice is motivated by the fact that CoT reasoning has proven to be most effective in this domain (Sprague et al., 2025), and most existing reasoning models are also benchmarked under mathematical scenarios. Second, our evaluation is conducted on relatively basic mathematical problems, rather than more challenging datasets such as MATH or AIME. On one hand, our goal in this paper is to demonstrate the effectiveness and potential of MARCOS as a latent thinking paradigm, rather than pursuing leaderboard-oriented results. On the other hand, tackling complex mathematical problems may require additional techniques such as reinforcement learning, which we leave

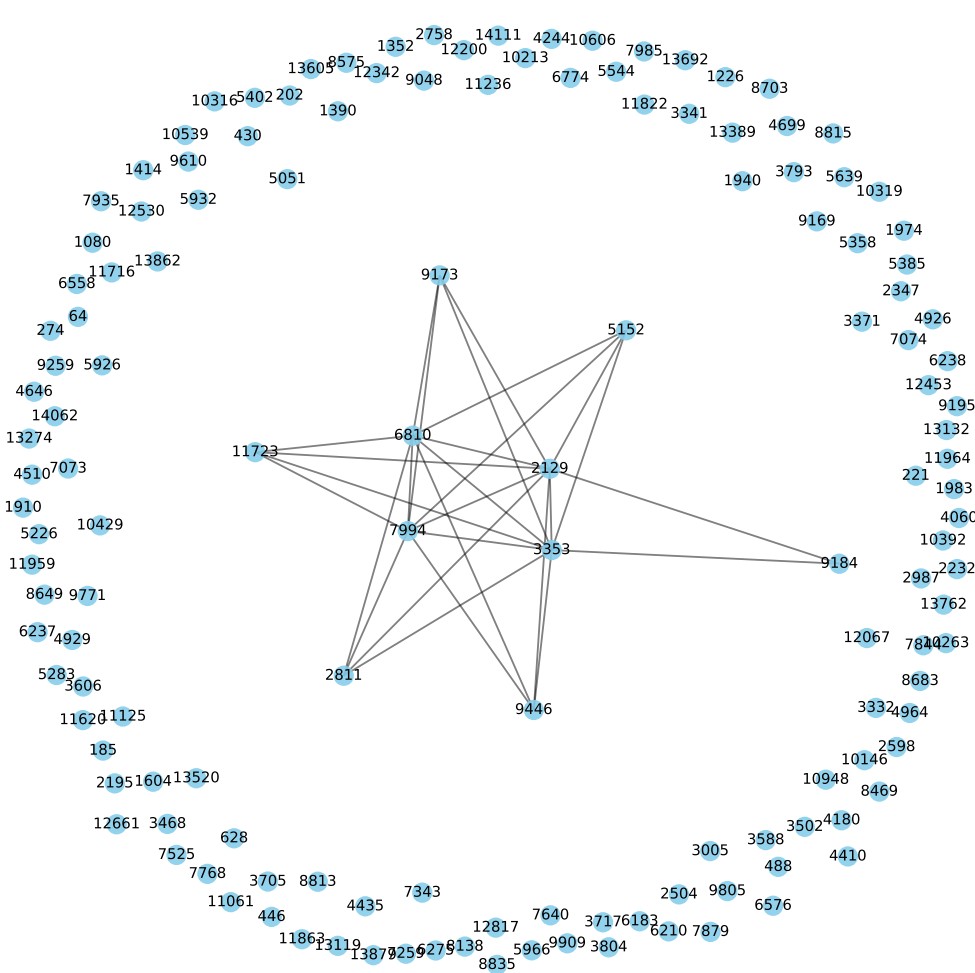

Figure 8: Correlation graph of the top 1% most frequently activated dimensions in $\boldsymbol{R}_k$.

for future work. Third, as discussed in Section 4.1, latent thinking represents a new paradigm for LLMs. To fully realize its potential, we need to explore corresponding pretraining methods and data, enabling the model to acquire broader and more robust capabilities across diverse domains.

Beyond these limitations, our study also opens several avenues for future work. First, and most importantly, is to develop a pre-training methodology for MARCOS. The key challenge may be the definition and segmentation of reasoning steps, as such large, unstructured corpora lack explicit, step-by-step annotations. One potential solution is to treat each sentence as a single reasoning step. Under this setup, the pre-training objective can be formulated as given the preceding sentences (analogous to the question in our paper), the model needs to generate the subsequent $K$ sentences. It is compatible with our existing training scheme, which could be scaled up. Second, as validated in Section 4.4, our random variable enables step-level control over stochasticity, offering a new perspective for exploration in reinforcement learning. Unlike conventional token-level sampling, our approach allows sampling reasoning pathways at a higher level, potentially making exploration more efficient. Third, our framework shows that randomness can be simulated "once" rather than being iteratively modeled, which may inspire new directions for diffusion language models. Last but not least, our work demonstrates the feasibility of disentangling thinking from speaking, suggests that speaking can be treated as a pure translation process. Therefore, it is worth trying to introduce more efficient decoding techniques, especially NAR decoding, to further accelerate inference.

## E    STATEMENT ON THE USE OF LLMS

We used LLMs to polish and improve the clarity, grammar, and fluency of our paper. The LLMs were not involved in any technical idealizations, derivations, or generation of research content. We have carefully reviewed all content generated by LLMs to ensure that no factual or technical inaccuracies were introduced, and the scientific integrity of the work is fully maintained.

