# OpenReview forum: "MARCOS: Deep Thinking by Markov Chain of Continuous Thoughts"
_ICLR.cc/2026/Conference — Submitted to ICLR 2026_

### Official Review · Reviewer_yNrV · 2025-10-29

**Soundness:** 1
**Presentation:** 1
**Contribution:** 1
**Rating:** 2
**Confidence:** 2

**Summary:**

MARCOS is a new method for continuous (latent) reasoning, as opposed to token-based chain-of-thought (CoT). It reasons in a latent space of neurons (deep and shallow) for K steps, with iterative transitions of continuous thoughts with incorporated randomness. It reports results superior to CoT, and other continuous reasoning methods in a 'from-scratch' training setting (see weaknesses).

**Strengths:**

The novelty of the method is a major strength. It is quite different from other continuous reasoning methods proposed, such as iCoT,  Coconut or CoLaR: as said above it reasons in a latent space of neurons (deep and shallow) for K steps, rather than being autoregressively similar to token generation (while using continuous vectors). Results in the setting provided are good (but see weaknesses).

**Weaknesses:**

The GSM8k accuracy numbers seem very low? I mean for all methods, including the baselines. Even in the COCONUT paper (which is relatively old by now) they reached 43% for the baseline, and 34% for COCONUT, although I understand those aren’t based on text (equations)? Shouldn’t you start with modern Qwen baselines that are much better than this?
I think 3B models with CoT should be in the 80-90% range (not ~13%)?  It also looks very strange that GSM8k doesn’t improve from 0.5B -> 3B?

When you say trained from scratch, do you mean with only the GSM8k data, or with pretraining data? The whole point of, and performance gains from CoT, I believe are because it takes advantage of strong pretraining data, which includes human thoughts. I believe this is why it’s hard to match with continuous thoughts, unless we figure out how to match that with a similar principle or idea…  Therefore, not comparing to standard methods of training here seems to be not a good comparison.

Small things:

Very first word of first sentence of intro:  “large” -> “Large”

“leaving little room for thoughtful planning beforehand” – I think CoT has been shown to provide good planning no? And the citations are pre-CoT (1993)?

Generally the method description is hard to follow (which is always hard on something fundamentally different, but still, extra care must be made to make it super clear I think).

**Questions:**

See weaknesses.

---

> ### Author Response · Authors · 2025-11-18
>
> Thank you for reviewing our paper! The following are our responses.
>
> $\bf{Q1}$: The GSM8k accuracy numbers seem very low? Shouldn’t you start with modern Qwen baselines that are much better than this? It also looks very strange that GSM8k doesn’t improve from 0.5B -> 3B?
>
> $\bf{A1}$: Thank you for your valuable questions. As we explained in our Common Response 1, the disentangled structure and stochastic continuous mapping of MARCOS makes it so different from existing token-based CoT models, and as such, it is not directly compatible with initializing from a pretrained LLM checkpoint. Therefore, **for fair comparison, all baselines were trained from scratch**. To be specific, the training process for these baselines starts by fine-tuning a CoT model from a randomly initialized checkpoint, and then using this model to train the baseline (both on the GSM8K-aug dataset).
>
> To address your concerns, we further reran the baselines using the pretrained Qwen2.5-0.5B checkpoint for initialization. For comparison, we also initialized the Understander and Speaker in our MARCOS using the pretrained Qwen2.5-0.5B. As shown in Common Response 1, the baseline results are very similar to other papers (e.g., [1]), with minor differences due to the use of a different backbone. This confirms that our implementation is correct. Furthermore, under these conditions, the performance of our model is still significantly superior!
>
> Currently, We are pretraining MARCOS and baselines on at least 1T tokens now, and will release the results as soon as possible.
> For the results of 3B models, the training dataset size may not be large enough to fully reflect scaling laws in the context of these larger models. The inclusion of both 3B and 0.5B parameter models in our experiments was intended to ensure a sufficient comparison, as we wanted to avoid potential bias due to differences in model size, which could lead to unfair results.
>
> [1] Think Silently, Think Fast: Dynamic Latent Compression of LLM Reasoning Chains, NeurIPS, 2025.
>
> $\bf{Q2}$: Do you mean with only the GSM8k data, or with pretraining data? Not comparing to standard methods of training here seems to be not a good comparison.
>
> $\bf{A2}$: By "trained from scratch," we mean that the models were only trained on the GSM8K-aug dataset without any prior pretraining. We fully agree that pretraining is an important next step. We are actively working on this, but due to the large amount of data and computational resources required for pretraining, we are not yet able to report those results. In the current paper, since all models (including both token-based CoT and continuous reasoning baselines) share the same training setup, we believe our comparisons at least show that MARCOS works well on medium-sized data and has the potential to beat large LLMs when pretrained on large-scale datasets.
>
> $\bf{Q3}$: I think CoT has been shown to provide good planning no? And the citations are pre-CoT (1993)?
>
> $\bf{A3}$: Apologies for the confusion caused. While token-by-token methods might still perform well on simpler tasks, there is growing evidence that large models struggle with planning for more complex problems[1,2,3,4]. The citation you referred to (1993), in fact, discusses human planning abilities, but we will update this to better cite current discussions on LLMs' planning capabilities.
>
> [1] On the planning abilities of large language models-a critical investigation
>
> [2] Llm+ p: Empowering large language models with optimal planning proficiency
>
> [3] Position: LLMs can’t plan, but can help planning in LLM-modulo frameworks
>
> [4] Large language models still can't plan (a benchmark for LLMs on planning and reasoning about change)
>
> $\bf{Q4}$: Generally the method description is hard to follow and some typos.
>
> $\bf{A4}$: Thanks for your detailed review. We will make further efforts to incorporate all your reviews and polish the explanation in the manuscript to ensure it is more clear and accessible. Besides, we will carefully review and correct the typos in the revised version.

---

> > ### Author Response · Authors · 2025-11-26
> >
> > Dear Reviewer,
> >
> > Thank you very much for your time and expertise in reviewing our submission. As the rebuttal deadline is approaching, please let us know if you have any questions or need any clarification. We are very happy to discuss any aspect of the work

---

### Official Review · Reviewer_3HPJ · 2025-10-31

**Soundness:** 2
**Presentation:** 3
**Contribution:** 2
**Rating:** 2
**Confidence:** 3

**Summary:**

This paper proposes MARCOS (Markov Chain of Continuous Thoughts), a novel paradigm for reasoning in LLMs that models the reasoning process as a hidden Markov chain over continuous representations rather than discrete token sequences. The key innovation is separating transitions between continuous thought states, or the "thinking" process from "speaking" (translating thoughts to natural language), with explicit modeling of randomness via variational modules. A variational module introduces a step-level random variable to control stochasticity and allow interpretable knobs over depth, verbosity, and format. The method is evaluated on mathematical reasoning benchmarks (GSM8K, SVAMP, MultiArith), achieving performance comparable to or exceeding token-based Chain-of-Thought with significant speedups.

**Strengths:**

1. The explicit separation of thinking and speaking tokens with use of variational modeling of randomness is a novel blend of ideas, and the $R_k$ module gives step-level control of randomness.
2. MARCOS has efficiency gains, reporting significant speedups, and show that non-autoregressive speaking still holds up reasonably.
3. Ablations are comprehensive and show how each component matters.

**Weaknesses:**

1. The main limitation is that all models were trained from scratch on the GSM8k-Aug dataset, making it difficult to quantify how MARCOS would perform against a pretrained LLM. It is unclear whether this paradigm is compatible with large-scale pretraining or if it only works when trained from scratch on a specific task.
2. The evaluation is confined to arithmetic word problems. These tasks are highly structured and the steps of reasoning are well-defined. Moreover, the mathematical reasoning tasks tested on are also relatively simple tasks, so MARCOS does not establish competitiveness on harder tasks such as MATH and AIME.
3. The approach depends heavily on $R_k$ sparsity, leading to model collapse without it, yet provides limited theoretical justification for why this bias yields stable, disentangled "factors" of reasoning.

**Questions:**

1. The approach currently uses a fixed number of reasoning steps, K=3. How would a dynamic K be potentially implemented as this is crucial for problems requiring more reasoning steps?
2. Have you tried mixture-of-Gaussians or flows for $R_k$​? If not, what evidence indicates the Gaussian prior with nonlinear mapping suffices to capture multi-modal path distributions?
3. The train from scratch evaluation makes it difficult to compare against the standard practice of finetuning large pretrained models . Have you experimented with initializing the "Understander" and "Speaker" modules from a pretrained model (e.g., Qwen2.5-0.5B) and only training the "Thinker" and variational components? If so, how did this impact performance and training stability?

---

> ### Author Response · Authors · 2025-11-18
>
> Thanks for your questions and suggestions! The following are our responses.
>
> $\bf{Q1}$: Difficult to quantify how MARCOS would perform against a pretrained LLM?
>
> $\bf{A1}$: Thank you for your insightful questions. As mentioned in our Common Response 1, our architecture differs significantly from existing token-based CoT models, and as such, it is not directly compatible with initializing from a pretrained LLM checkpoint. However, this does not imply that large-scale pretraining is not possible for our model. We are indeed exploring this direction, but due to the substantial computational resources and data required, we are currently unable to report those results. Importantly, the efficiency advantages of our approach are independent of whether pretraining is used, so we are confident in the potential of our architecture, and we believe this innovative aspect of the model is valuable for ICLR.
>
> $\bf{Q2}$: The evaluation is confined to arithmetic word problems.
>
> $\bf{A2}$: Thank you for this valuable suggestion. We agree that testing on more complex datasets would further validate the model. As we emphasize in our paper, the primary contribution is the introduction of a new architecture, which we’ve validated on smaller, well-structured datasets like GSM8K. These benchmarks are widely used for comparison in the field, and many continuous reasoning works are also evaluated on them [1,2,3]. While this does not yet establish MARCOS as competitive on more difficult tasks like MATH and AIME, it demonstrates that our model has the potential to outperform existing methods in terms of inference speed and reasoning accuracy. We believe that achieving strong performance on these more complex tasks would require pretraining and reinforcement learning, which are currently outside the scope of this work. This paper is a starting point, and we plan to explore these challenging benchmarks and incorporate other techniques in future work.
>
> [1]Training Large Language Models to Reason in a Continuous Latent Space
>
> [2]Implicit Chain of Thought Reasoning via Knowledge Distillation
>
> [3]Think Silently, Think Fast: Dynamic Latent Compression of LLM Reasoning Chains
>
> $\bf{Q3}$: Theoretical justification for why the bias of sparsity yields stable, disentangled "factors" of reasoning.
>
> $\bf{A3}$: We have a thorough explanation on this. Please refer to Common Response 2.
>
> $\bf{Q4}$: How would a dynamic $K$ be potentially implemented
>
> $\bf{A4}$: Thanks for your thoughtful question. In our current experiments, $K=3$ has proven to be a suitable choice across the three benchmarks. However, it’s important to clarify that $K=3$ does not limit the solution to exactly three steps. For example, in some cases (e.g., case 1, column 3 in Figure 6), our model may take two steps within the first reasoning step. Therefore, $K=3$ serves as a fast validation setting. For more complex problems, we think that it may be difficult to adjust $K$ manually, because we cannot predict the difficulty of a problem before reasoning starts. Nevertheless, our model could potentially scale up, allowing us to train with $K=3$ and then test with a higher value like $K=5$. Additionally, we could introduce another classifier module to determine whether a given step is the final one, which we are very willing to explore.
>
> $\bf{Q5}$: Mixture-of-Gaussians or flows for $R_k$?
>
> $\bf{A5}$: Thank you for this insightful suggestion. We agree that exploring mixture-of-Gaussians or flow-based models for $R_k$ could be valuable and is an interesting direction for future work. However, this does not impact the core contribution of our current paper. Introducing such models could potentially require setting a prior for the number of components in the mixture, which might add additional complexity. At this stage, our focus has been on the foundational architecture of MARCOS, and the current Gaussian approach is a simple yet effective starting point. We plan to investigate such alternatives as part of future work.
>
> $\bf{Q6}$: Have you experimented with initializing the "Understander" and "Speaker" modules from a pretrained model.
>
> $\bf{A6}$: Thank you for your constructive suggestion. Following your comments, we have supplemented an experiment. Please refer to the results in Common Response 1.

---

> > ### Author Response · Authors · 2025-11-26
> >
> > Dear Reviewer,
> >
> > Thank you very much for your time and expertise in reviewing our submission. As the rebuttal deadline is approaching, please let us know if you have any questions or need any clarification. We are very happy to discuss any aspect of the work

---

### Official Review · Reviewer_PWgj · 2025-11-01

**Soundness:** 3
**Presentation:** 2
**Contribution:** 4
**Rating:** 6
**Confidence:** 2

**Summary:**

This paper decouples an LM’s internal and external thought processes by casting them as the latent and observed variables of a conditional Hidden Markov Model (cHMM). The authors use supervised CoT training to learn weights that maximize data likelihood via a summed ELBO—one term per supervised time step in the cHMM. They compare favorably to other supervised CoT methods, including Coconut and CoLaR.

**Strengths:**

* The theoretical setup is clean and helps justify what might otherwise feel like an unusual number of training components.
* Results are favorable relative to CoLaR, which is already a strong supervised baseline.
* All results are pre-RL, so there may be further headroom.

**Weaknesses:**

* Parts of the presentation are unclear, with some important details missing (see Question).
* I would prefer more evaluations in place of Figures 4–6, which focus on interpretability that seems tangential. In particular, please investigate why the sparsity penalty is so crucial—it feels like an outlier in an otherwise elegant design. Would an entropy-based penalty perform similarly?

**Questions:**

What exactly drives the reported 15.7× inference speedup? Even if intermediate text is not always emitted, generation still proceeds stepwise, so I would expect comparable latency unless speaking is largely bypassed. From the appendix I see K=3, but how many tokens did CoT-SFT produce on average? Are latent steps more stepwise-efficient than CoT tokens, or is the speedup primarily from answer-only decoding?

---

> ### Author Response · Authors · 2025-11-18
>
> We sincerely appreciate your recognition of the significance of our contribution, the clarity of our setups, the favorable evaluation of our experimental results, and the potential for future improvements.
>
> $\bf{Q1}$: What exactly drives the reported 15.7× inference speedup? How many tokens did CoT-SFT produce on average? Are latent steps more stepwise-efficient than CoT tokens, or is the speedup primarily from answer-only decoding?
>
> $\bf{A1}$: Thank you for your insightful and important questions. To address your concerns, we need to conduct a more detailed analysis of our model's efficiency.
>
> In the `thinking` stage, the model needs to perform $K$ steps of reasoning sequentially, and this is a necessary process that cannot be skipped. However, this is still faster than token-based step-by-step reasoning because we are performing computations in continuous space. In the `speaking` stage, we have two options: we can either decode all steps in parallel or decode only the final step to generate the answer. In practice, both methods result in similar efficiency, as the latency primarily comes from the decoding of a single step, which is the bottleneck in both cases.
>
> Thus, the reported 15.7× inference speedup is driven by the combination of efficient multi-step continuous reasoning in the thinking phase and single-step decoding in the speaking phase. Both of these components are crucial for achieving the speedup. It is difficult to measure which part contributes more, as both are essential components of the model and cannot be separately ablated.
>
> To provide further context, for the CoT-SFT, the average numbers of tokens produced are: 90.80/59.05/62.23 for GSM8K, SVAMP, MultiArith, respectively.
>
> $\bf{Q2}$: Why is the sparsity penalty so crucial?
>
> $\bf{A2}$: Please refer to Common Response 2.

---

> > ### Author Response · Authors · 2025-11-26
> >
> > Dear Reviewer,
> >
> > Thank you very much for your time and expertise in reviewing our submission. As the rebuttal deadline is approaching, please let us know if you have any questions or need any clarification. We are very happy to discuss any aspect of the work

---

### Official Review · Reviewer_ifwg · 2025-11-07

**Soundness:** 1
**Presentation:** 2
**Contribution:** 2
**Rating:** 2
**Confidence:** 4

**Summary:**

This paper introduces MARCOS, a new reasoning paradigm for large language models. The goal is to replace token-based Chain-of-Thought (CoT) with a continuous-space model. The authors propose modeling reasoning as a "conditional hidden Markov model (cHMM)" where latent, high-dimensional "thoughts" transition independently from the "speaking" process of generating tokens. The model is trained with a two-phase variational scheme. The authors claim this approach achieves performance comparable to or exceeding token-based CoT and other continuous reasoning methods, while being significantly faster.

**Strengths:**

1. Well-Motivated Problem: The paper correctly identifies the limitations of autoregressive, token-by-token CoT reasoning: it is slow, entangles reasoning with token generation, and creates an information bottleneck. The goal of decoupling thinking and speaking is a valuable research direction.

2. Novel Architecture: The core idea of a dual-component latent space ($Neu^{deep}$ and $Neu^{shallow}$) and the use of a variational framework to manage randomness ($R_k$) at the step level rather than the token level is interesting.

3. Analysis of Randomness: The analysis in Section 4.4, which attempts to show how different dimensions of the random variable $R_k$ control distinct properties like sentence length and format, is a good piece of analysis and a promising direction for interpretable control.

**Weaknesses:**

1. Inverted scaling in Table 1:
A critical issue is the evidence of inverted scaling in Table 1. For nearly all continuous baselines (Coconut, CoLaR, CODI), the 3B parameter models perform significantly and anomalously worse than their 0.5B counterparts. This inverted scaling is not expected and suggests a deep issue with the training or implementation of these larger baselines, rendering them invalid as good-faith comparison points.

2. Baseline performance:
 The reported scores for baselines like Coconut seem significantly lower than those reported in their original papers.

3. The paper introduces a large-scale, complex framework (cHMM, randomness encoders/predictors, $Neu^{deep/shallow}$) without adequately justifying why this specific formalism is necessary. It is not clear why a simpler recurrent state-space model would not suffice. The paper is heavy on new jargon, making it difficult to position this work relative to the vast existing literature on VAEs, latent variable models, and structured prediction.

**Questions:**

1. In Table 1 ablations, without sparsity loss, the model performance drops dramatically. It makes me wonder the necessity of ELBO formulation of the problem. Also it's unclear whether there exists the issue of posterior collapse. To me, the model's success is an artifact of sparsity prior rather than variational framework itself.

---

> ### Author Response · Authors · 2025-11-18
>
> Thanks a lot for your questions and suggestions!
>
> $\textbf{Q1}$: Inverted scaling in Table 1.
>
> $\bf{A1}$: Thank you for your detailed observation. As explained in our Common Response 1, all baselines were randomly initialized and trained using the same training set, GSM8K-aug, which contains 385K samples. This dataset size may not be large enough to fully reflect scaling laws in the context of these larger models. The inclusion of both 3B and 0.5B parameter models in our experiments was intended to ensure a sufficient comparison, as we wanted to avoid potential bias due to differences in model size, which could lead to unfair results.
>
> $\bf{Q2}$: Baseline performance.
>
> $\bf{A2}$: We appreciate your attention to the differences in baseline performance. The baselines in our paper were implemented by only modifying the initialization in their source code. Specifically, the training process for these baselines starts by fine-tuning a CoT model from an initialized checkpoint (denoted as A), and then using this model to train the baseline. The only difference in our implementation is that, while the original code uses a pretrained Qwen checkpoint for A, we used a randomly initialized checkpoint. As a result, the differences you observed are expected.
>
> To address your concerns, we further reran the baselines using the pretrained Qwen2.5-0.5B checkpoint for A. For comparison, we also initialized the Understander and Speaker in our MARCOS using the pretrained Qwen2.5-0.5B. As shown in Common Response 1, the baseline results are very similar to other papers (e.g., [1]), with minor differences due to the use of a different backbone. This confirms that our implementation is correct. Furthermore, under these conditions, the performance of our model is still significantly superior!
>
> We are currently working on pretraining MARCOS, but due to the large data and compute requirements, we are unable to report that result yet. However, we believe that the contributions made in this paper meet the standards for ICLR.
>
> [1] Think Silently, Think Fast: Dynamic Latent Compression of LLM Reasoning Chains, NeurIPS, 2025.
>
> $\bf{Q3}$: Justify why the model formalism is necessary.
>
> $\bf{A3}$: Regarding the need for our model's design,
>
> -  Our primary goal was to enable continuous reasoning. However, as the output must be discrete tokens (natural language), it is necessary to introduce a “translator” to translate the continuous thinking to discrete tokens. Therefore, we propose separating the `thinking` and `speaking` stages, where thinking is treated as updating the continuous thoughts (i.e., hidden state in a Markov chain), while speaking is to obtain the tokens (i.e., observable variables).
> -  The need for the $Neu^{deep}$ and $Neu^{shallow}$ components comes from inspiration drawn from how the human brain categorizes thoughts, and this was validated through our ablation study in Section 4.3.
> -  The randomness predictor is critical because we know the data contains inherent randomness. Traditional token-sampling randomness would create a mixed, noisy continuous thought, as it would need to cover all possible decoded outputs. To resolve this, we shifted the randomness to the thinking stage and used a randomness predictor to sample it in inference time.
> -  The randomness encoder is required to handle the posterior over randomness when training with the ELBO formulation. So it is a necessary component from a design standpoint.
>
> $\bf{Q4}$: The model's success is an artifact of sparsity prior.
>
> $\bf{A4}$: Please refer to Common Response 2.

---

> > ### Author Response · Authors · 2025-11-26
> >
> > Dear Reviewer,
> >
> > Thank you very much for your time and expertise in reviewing our submission. As the rebuttal deadline is approaching, please let us know if you have any questions or need any clarification. We are very happy to discuss any aspect of the work

---

### Author Response · Authors · 2025-11-18
**Common Reasponse [1/2]**

Dear AC and all reviewers,

Thank you all for taking the time reviewing our paper! We are particularly encouraged by the reviewers’ recognition of the following strengths of our paper:
- **Novelty**: “novel architecture” (Reviewer ifwg), “theoretical setup is clean” (Reviewer PWgj), “novel paradigm, novel blend of ideas” (Reviewer 3HPJ), “novelty is a major strength” (Reviewer yNrV).

- **Effectiveness**: “good piece of analysis and promising direction for interpretable control” (Reviewer ifwg), “results are favorable” (Reviewer PWgj), “efficiency gains, significant speedups” (Reviewer 3HPJ), “ablations are comprehensive” (Reviewer 3HPJ), “results are good” (Reviewer yNrV).

- **Significance**: “well-motivated, identifies the limitations of autoregressive, token-by-token CoT reasoning” (Reviewer ifwg), “a valuable research direction” (Reviewer ifwg), “further headroom” (Reviewer PWgj).

We’ve noticed that although all reviewers think the proposed method (MARCOS) is novel, effective, and significant, 3 out of 4 reviewers gave the score "2" as the overall rating. The most critical argument is that `we did not pretrain MARCOS to compare it with stronger baselines that are pretrained on massive data`. Another problem is that `in our ablation, removing the sparsity loss can cause dramatic decrease in performance, so it looks like other parts of the model are not necessary`. In this general response, we will try to explain why we do not have results of pretrained models, and why sparsity is so crucial in the training of MARCOS.

**Common Response 1: Compare MARCOS with Pretrained Baselines**

MARCOS has a very different structure from existing reasoning models. Unlike traditional CoT, where LLMs reason by progressively generating discrete tokens, MARCOS thinks in a high-dimensional continuous space. To do so, we disentangle `thinking` and `speaking` processes in traditional LLMs, where MARCOS first thinks in continuous space and then translates the latent reasoning step to natural language. Consequently, MARCOS can think freely without being restricted by the information bottleneck caused by discretization. Meanwhile, `speaking` becomes optional. Like a real human, MARCOS can think silently, and only decode intermediate thoughts when asked to. This leads to up to 15x speed up and higher reasoning accuracy simultaneously compared with traditional LLMs as shown in Table 1.

Though these significant structural innovations can bring exciting benefits, they also make MARCOS incompatible with the existing LLM structure. The main difference is that MARCOS is learning a stochastic high-dimensional continuous mapping, instead of mappings between discrete tokens. We have tried to initialize MARCOS with weights from pretrained LLMs, such as Qwen-2.5, but found that training got stuck quickly at local minimums. For other baselines compared in the paper, they are essentially compressing the CoT tokens into continuous representations, making it possible to gradually transform a token-based LLM to them, which is not the case for MARCOS. As a result, **we train MARCOS from scratch** on medium-sized datasets as a verification of the concept, which has shonw the potential of MARCOS.

We fully agree that large-scale pre-training is needed to show that MARCOS also works well on scale, and could compete with SOTA token-based LLMs in practice. In fact, we have been working on the pre-training of MARCOS since a month ago, aiming to train it on at least 1T tokens, which will take about 4 more months given our current computation power.

But to address these concerns, inspired by Reviewer 3HPJ, we supplement an experiment by initializing the Understander and Speaker in our MARCOS using the pretrained Qwen2.5-0.5B. The baselines are also initialized with this checkpoint. It is important to note that the results of baselines can vary due to differences in the backbone used in different papers. For example, in [1], the accuracy of Coconut on GSM8K is 34%, while in [2], after switching to Llama3.2, the accuracy drops to 23%. As shown in the table below, our results are similar to those in [2], confirming that our implementation is correct. Furthermore, under these conditions, the performance of our model is still significantly superior!
| | GSM8K|SVAMP|MultiArith|
|-|-|-|-|
|iCoT-SI |13.9|47.3|30.5|
|coconut|23.6|50.8|46.7|
|CoLaR|17.2|40.3|48.3|
|MARCOS|24.1|48.5|49.2|
|MARCOS(text)|26.6|57.2|48.7|

As a result, we believe MARCOS opens up a new paradigm for complex reasoning. which should be of interest to researchers and engineers working on LLM reasoning and structure.

[1]Training Large Language Models to Reason in a Continuous Latent Space

[2]Think Silently, Think Fast: Dynamic Latent Compression of LLM Reasoning Chains

---

> ### Author Response · Authors · 2025-11-18
> **Common Response [2/2]**
>
> **Common Response 2: The Importance of Sparsity in the Training of MARCOS**
>
> There is an ablation study in Table 1 of our paper, showing a significant drop in accuracy when removing the sparsity constraint during training. Two reviewers are then concerned that the whole training paradigm is unnecessary except for the sparsity loss, which we believe is a severe misunderstanding.
>
> The training of MARCOS can be viewed as a conditional variational autoencoder (cVAE). For each reasoning step, MARCOS predicts the next step conditional on information from the question as well as thoughts at the current step. We find that even given all previous information, there remains some randomness in predicting the next step, regarding its reasoning direction, length, and expression. To accurately learn this remaining stochasticity, we follow the basic idea of cVAE to reproduce each step from example solutions as the posterior in ELBO during model training, while also aligning the prior and posterior distributions.
>
> We introduced the sparsity loss, because there are two information sources when reproducing each ground truth reasoning step, i.e., the condition (question and previous thoughts) and the ground truth reasoning step itself (i.e., the posterior explained above). We expect the model to mostly use information from the condition, and only extract necessary features from the ground-truth step to control the remaining stochasticity. As a result, we need to restrict the information bandwidth from the second information source. Otherwise, the model will easily find a shortcut from the ground-truth step to itself, while ignoring the condition. Our sparsity loss is used to do this. Therefore, **sparsity is a necessary condition for the model’s operation, rather than a factor in improving performance**. This design is not a specific choice among various priors (such as different priors mentioned by Reviewer ifwg and 3HPJ) or an elegant design (Reviewer PWgj), but rather an essential requirement for the model's functionality.
>
> We agree with the reviewers that the sparsity loss is critical to the performance of MARCOS. However, it is a part of the training algorithm, whose necessity does not indicate that all other parts are redundant.

---

### Author Response · Authors · 2025-12-01
**Summary of Rebuttal [1/2]**

Dear Program Chairs, Senior Area Chairs, Area Chairs and Reviewers,

We sincerely thank all reviewers for recognizing the **novelty** of our MARCOS, the **effectiveness** of our experiments (both in terms of accuracy and efficiency gains), and the **strong potential** of this new reasoning paradigm. Unfortunately, no reviewers responded during the rebuttal period, but we summarize how we have addressed their concerns as follows:

***

### Common Concerns

**1. Baseline Performance.**

As we explained in the rebuttal, MARCOS’s architecture, with disentangled continuous thoughts vs. speaking and step-level randomness in a cHMM / conditional VAE formulation, is structurally different from token-based CoT models. Therefore, it cannot be trivially initialized from off-the-shelf pretrained LLM checkpoints that are designed for token-by-token reasoning. For fair comparison, in our main experiments, all continuous baselines were therefore trained from scratch under the same data and training protocol, which naturally leads to accuracy numbers different from those reported in works that do rely on large-scale pretraining. To further address this concern, we added experiments where both MARCOS (its Understander/Speaker) and all baselines are initialized from the same pretrained Qwen2.5-0.5B checkpoint; in this setting, baseline results closely match prior work and MARCOS still outperforms them, confirming that the discrepancy is due to different initialization regimes rather than implementation issues.

**2. Role of Sparsity.**

Regarding sparsity, at each reasoning step MARCOS can be viewed as a conditional VAE with two information sources: the condition (question + previous thoughts) and the ground-truth next step. Without any restriction, the model can take a shortcut by encoding the entire next step into the latent, ignoring the condition. Our sparsity loss limits the information bandwidth of the latent, forcing the model to use the condition for the main logical content and to use the ground truth only for training. The large performance drop when removing sparsity is therefore expected: sparsity is a necessary part of the training objective that makes the conditional VAE formulation work as intended, not an incidental trick that replaces the rest of the framework.

***

### Summary for Reviewer ifwg:

•	**W1: Inverted scaling and low baseline performance.**

All baselines were initially trained from scratch on GSM8K-aug (385K samples), which is too small to reliably exhibit scaling laws for 3B models, causing inapparent scaling. To clarify, we re-ran baselines and MARCOS with pretrained Qwen2.5-0.5B initialization; under this setting, the baselines match prior papers and MARCOS remains clearly superior.

•	**W2: Baseline implementation correctness.**

The only difference from original code is the initialization (random vs. pretrained). With Qwen2.5-0.5B initialization, our baseline results align with existing reports, confirming that our implementation is correct.

•	**W3: Necessity of the formalism vs. simpler models.**

The cHMM / conditional VAE structure, with separated thinking/speaking and explicit randomness, is required to (i) support continuous reasoning with discrete outputs, and (ii) move stochasticity into the latent thinking space instead of noisy token sampling. Ablations show that each component (thought decomposition, randomness encoder/predictor) is needed for stable and effective training.

•	**W4: Success is just sparsity prior.**

We clarify that sparsity is a structural requirement to prevent shortcut solutions and ensure the latent captures only residual stochasticity, as detailed above; it complements, rather than replaces, the variational framework.

***

### Summary for Reviewer PWgj

•	**W1: Source of the 15.7× speedup.**

MARCOS performs K steps of reasoning purely in continuous space, then decodes only once (all steps in parallel or just the final step). CoT-SFT, by contrast, generates steps sequentially. This combination of continuous multi-step reasoning + single decoding step yields our reported up to 15.7× inference speedup.

•	**W2: Why sparsity is so crucial / whether a different prior would suffice.**

The strong dependence on sparsity is expected from the conditional VAE perspective: without sparsity, the latent can memorize the next step and ignore the condition. Sparsity enforces the intended factorization (condition for main logic, latent for residual stochasticity). Other priors (e.g., entropy penalties or alternative distributions) are interesting directions but orthogonal to the core contribution of MARCOS.

---

> ### Author Response · Authors · 2025-12-01
> **Summary of Rebuttal [2/2]**
>
> ### Summary for Reviewer 3HPJ
>
> •	**W1: Difficult to quantify performance vs. large pretrained LLMs.**
>
> Because MARCOS’s architecture is incompatible with direct LLM checkpoint loading, we first validate it in a from-scratch setting where all methods share identical training data and protocol. We then added experiments where Understander/Speaker and baselines are initialized from Qwen2.5-0.5B; MARCOS still outperforms continuous baselines, showing strong potential even in a pretrained regime. Full large-scale pretraining (≥1T tokens) is ongoing but beyond the rebuttal timeline.
>
> •	**W2: Evaluation confined to arithmetic word problems.**
>
> Our primary goal is to introduce and validate a new reasoning paradigm. We therefore follow established practice and evaluate on standard math word problem benchmarks (GSM8K, SVAMP, MultiArith), which are also used by related continuous reasoning works. We agree that extending to harder datasets like MATH and AIME is important and plan to explore them with pretraining and RL in future work.
>
> •	**W3: Why sparsity leads to stable, disentangled “factors”.**
>
> As discussed in the common concerns, sparsity prevents shortcut memorization and encourages the latent to capture structured residual variations (e.g., verbosity, format), which is exactly what we observe empirically in our analyses.
>
> •	**W4: Dynamic K, richer priors (mixture-of-Gaussians/flows).**
>
> We clarify that K=3 is a fast validation setting and does not strictly bound the logical steps; multiple micro-steps can be embedded in one latent step. MARCOS can in principle be trained with smaller K and tested with larger K, or extended with an additional classifier to decide when to stop. More expressive priors (mixtures, flows) are promising but would introduce extra complexity; we view them as natural follow-up work rather than essential to the core architecture.
>
> ***
>
> ### Summary for Reviewer yNrV
>
> •	**W1: Low GSM8K accuracies and lack of strong Qwen baselines.**
>
> All methods (CoT and continuous baselines) were trained from scratch on GSM8K-aug for fairness, which naturally yields lower accuracies than fully pretrained LLMs. After adding a Qwen2.5-0.5B-initialized setting, the baselines reach accuracies comparable to those in prior work, and MARCOS/MARCOS(text) still outperform them, showing that the method is competitive even under stronger backbones.
>
> •	**W2: “Trained from scratch” vs. pretraining.**
>
> “From scratch” means training only on GSM8K-aug without external pretraining. We agree that full pretraining is crucial for reaching SOTA numbers and are actively working on it; however, under a strictly controlled from-scratch setup, MARCOS already shows strong accuracy–efficiency trade-offs, indicating its potential when scaled up.
>
> •	**W3: CoT and planning, clarity and typos.**
>
> We clarify that while CoT can improve planning on simpler tasks, several recent works show limitations of LLM planning on more complex settings; we will update citations accordingly and revise the presentation/typos to make the method clearer and more accessible.

---

### Meta-Review · Area_Chair_gjFe · 2026-01-07

**Summary:**

This paper proposes MARCOS, a new continuous latent reasoning paradigm that decouples thinking (continuous latent transitions) from speaking (token generation), formulated as a conditional HMM / conditional VAE with step-level randomness and sparsity constraints. Reviewers broadly agree the idea is novel and intellectually interesting, with strong inference-efficiency gains and thoughtful analysis of latent stochasticity. However, the evaluation and comparisons raised serious concerns, especially regarding from-scratch training only, very low absolute accuracies, inverted scaling behavior, and the heavy reliance on sparsity for performance. While the rebuttal adds clarifications and a small pretrained-initialization experiment, the core concerns about realism, scalability, and empirical grounding largely remain.

**Reviewer Concerns:**

Addressed or partially mitigated:
- Baseline correctness & inverted scaling: Authors clarified that baselines were trained from scratch on GSM8K-aug and showed that initializing all models from Qwen2.5-0.5B restores expected baseline behavior; MARCOS remains competitive in that controlled setting.
- Inference speedup explanation: The reported speedup is clarified as coming from continuous multi-step reasoning plus single-step (or parallel) decoding, with concrete CoT token counts provided.
- Role of sparsity: Authors give a coherent explanation that sparsity is structurally necessary to prevent posterior shortcuts in the conditional VAE, not an ad-hoc trick.

Still outstanding / major concerns:
- Lack of realistic comparison to modern pretrained LLMs: Most results rely on from-scratch training on GSM8K-aug, yielding very low absolute accuracies; this makes it hard to assess practical relevance or competitiveness.
- Scalability and generality: Evaluation is limited to relatively simple arithmetic word problems; no evidence is provided for harder reasoning tasks (e.g., MATH, AIME) or non-math domains.
- Heavy dependence on sparsity: Despite explanations, empirically the method collapses without sparsity, raising concerns that other components of the framework may be under-justified or redundant.
- Complexity vs. necessity: Several reviewers questioned whether the full cHMM / variational machinery is required, versus simpler latent or state-space models.
- Presentation clarity: Multiple reviewers found the method difficult to follow, with dense notation and heavy jargon, even after rebuttal clarifications.

**Reviewer Scores:**

Estimated post-discussion positions:
- ifwg: 2 → 2 (core concerns about scaling and sparsity remain).
- yNrV: 2 → 2 (fundamental objection to from-scratch setup and relevance to CoT remains).
- 3HPJ: 2 → 2–3 (novelty acknowledged, but evaluation limitations persist).
- PWgj: 6 → 6 (positive on theory and efficiency, but not decisive).
Overall, the reviewer pool remains strongly split, with multiple firm rejects.

---

### Decision · Program_Chairs · 2026-01-26

Reject